# Phylogenetic and Expression Analysis of SBP-Box Gene Family to Enhance Environmental Resilience and Productivity in *Camellia sinensis* cv. *Tie-guanyin*

**DOI:** 10.3390/plants14030422

**Published:** 2025-02-01

**Authors:** Yusen Gao, Yingxin Wen, Qinmin Lin, Yizhuo Feng, Xinying Shi, Siyao Xiao, Elisabeth Tumukunde, Kehui Zheng, Shijiang Cao

**Affiliations:** 1College of Mechanical and Electrical Engineering, Fujian Agriculture and Forestry University, Fuzhou 350002, China; gyusen1990@163.com (Y.G.); 18965982768@139.com (S.X.); 2College of Forestry, Fujian Agriculture and Forestry University, Fuzhou 350002, China; 18050409132@139.com (Y.W.); 15532106788@163.com (Y.F.); 3College of Life Sciences, Fujian Agriculture and Forestry University, Fuzhou 350002, China; 18760030926@163.com (Q.L.); 13163835793@163.com (X.S.); 4College of Chemical Engineering, College of Biological Science and Engineering, Fuzhou University, Fuzhou 350108, China; elizawkb22@gmail.com; 5College of Computer and Information Sciences, Fujian Agriculture and Forestry University, Fuzhou 350002, China

**Keywords:** tea, Tieguanyin, SBP, gene family, abiotic stress

## Abstract

Tieguanyin tea, a renowned oolong tea, is one of the ten most famous teas in China. The Squamosa Promoter Binding Protein (SBP)-box transcription factor family, widely present in plants, plays a crucial role in plant development, growth, and stress responses. In this study, we identify and analyze 22 *CsSBP* genes at the genome-wide level. These genes were distributed unevenly across 11 chromosomes. Using *Arabidopsis thaliana* and *Solanum lycopersicum* L. as model organisms, we constructed a phylogenetic tree to classify these genes into six distinct subfamilies. Collinearity analysis revealed 20 homologous gene pairs between *AtSBP* and *CsSBP*, 21 pairs between *SiSBP* and *CsSBP*, and 14 pairs between *OsSBP* and *CsSBP.* Cis-acting element analysis indicated that light-responsive elements were the most abundant among the *CsSBP* genes. Protein motif, domain, and gene architecture analyses demonstrated that members of the same subgroup shared similar exon–intron structures and motif arrangements. Furthermore, we evaluated the expression profiles of nine *CsSBP* genes under light, shade, and cold stress using qRT-PCR analysis. Notably, *CsSBP1, CsSBP17*, and *CsSBP19* were significantly upregulated under all three stresses. This study provides fundamental insights into the *CsSBP* gene family and offers a novel perspective on the mechanisms of SBP transcription factor-mediated stress responses, as well as Tieguanyin tea’s adaptation to environmental variations.

## 1. Introduction

Tieguanyin (*Camellia sinensis* var. *Tie-guanyin*), a traditional Chinese oolong tea, is one of China’s ten most famous teas [1]. Its flavor has a mellow sweetness with honey undertones, finishing cool and sweet for a distinct taste experience [2]. In addition to the general health benefits associated with tea consumption, it offers additional properties such as anti-aging effects, prevention of arteriosclerosis and diabetes, weight reduction, and detoxification from smoke and alcohol [3,4]. Due to the unique effect of Tieguanyin and the continuous promotion of tea culture [5], people’s requirements for tea are constantly increasing [6]. In recent years, tea plants in many regions of China have been subjected to various abiotic stresses, such as the sudden arrival of cold waves in early winter or unexpected cold stress and drought conditions during early spring [7,8]. These stresses repeatedly injure the tea trees, impacting the timing of harvest, yield, and quality of spring tea and causing serious economic losses [9]. To cope with the challenges brought by the extreme climate, it has become urgent to adapt to the adverse environmental conditions and improve the stress resistance of plants. Therefore, exploring the influence of various factors relevant to the growth and evolution of Tieguanyin is particularly important to enhance the quality and yield of tea [10].

As climate change and extreme weather events intensify, plants face a formidable set of challenges, including drought, extreme temperatures (both scorching heat and freezing cold), and intense light [11]. These stresses often interconnect, significantly influencing plant growth, development, and survival [12]. To cope with harsh environments, plants employ a variety of defense mechanisms to tolerate diverse unfavorable conditions [13]. In the context of higher plants, transcription factors (TFs) are DNA-binding proteins that selectively interact with specific sequences within the promoter regions of genes, either activating or repressing the transcription of multiple target genes [7,14,15]. Plants can initiate a complex defense system and adapt to abiotic stresses by modulating gene expression at the mRNA transcription level [16,17].

The Squamosa Promoter Binding Protein (SBP)-box transcription factors constitute a ubiquitous gene family within plants, performing essential functions in plant growth, development, and stress responses [18]. This gene family is characterized by a remarkably conserved SBP domain. One shared characteristic of SBP genes is that they encompass roughly 76 amino acid residues, two distinct zinc finger configurations, and a dual-component nuclear localization signal (NLS), which partially coincides with the second zinc finger pattern in the C-terminal segment [19,20]. The NLS sequence overlaps with the second zinc finger, facilitating nuclear transport and enabling transcriptional regulation of downstream genes [20]. The SBP domain encompasses four amino acid residues that coordinate zinc ions, critical for the protein’s structural integrity.

Previous research has demonstrated that the SBP gene is crucial for various plant functions. In many plants, such as Arabidopsis and silver birch, it can regulate leaf development and affect flower growth function [21,22], and in evergreen trees, such as apple and loquat, the SBP gene can control fruit ripening and manage hormone signaling [23,24,25]. Alternatively, SBP has a disease-responsive function and is also involved in glucose and ATP metabolism and in the transport of carbohydrates [26]. Moreover, several studies indicate that in many plant species, the SBP-box gene is also associated with the reaction to both biotic and abiotic stresses [27]. For instance, overexpressing VpSBP16 in *Arabidopsis thaliana* enhances tolerance to salt and drought stress during the processes of seed germination, as well as in the seedling and adult plant stages, through regulating the salt overly sensitive (SOS) pathway and reactive oxygen species (ROS) signaling cascades [19]. The abundance of stress-related cis-elements, especially those associated with drought and cold responses, in the upstream regulatory sequence of the *CsSBP* gene, combined with expression pattern analysis, strongly indicates an active role for *CsSBP* in facilitating the tea plant’s defense against abiotic stresses [28]. The SBP gene is frequently targeted by miR156, a microRNA that regulates its expression in response to various abiotic stress conditions [29]. For example, under high-temperature and low-nitrogen conditions, miR156 is upregulated, which in turn inhibits the expression of its target SBP gene [30,31]. Many stress response genes in foxtail millet (*Setaria italica*), especially *SISBP9, SiSBP10,* and *SISBP16*, are highly expressed under various stress conditions, which require further attention [32]. SBP genes are involved in interactions with hormonal signaling pathways, including those of abscisic acid (ABA) [33], salicylic acid (SA), and methyl jasmonate (MeJA) [34], thereby orchestrating plant physiological responses to biotic stressors. By regulating these hormonal pathways, SBP genes can amplify or inhibit hormonal activity, thus modulating the transcriptional regulation of genes associated with stress adaptation [35].

Advancements in next-generation sequencing technologies have facilitated the identification of SBP-box gene families across diverse plant species, with 16 SBP genes discovered in *Arabidopsis thaliana* so far [36], 14 in barley [37], 18 in grape [27], 8 in beet [38], 58 in oilseed rape [39], 27 in apple [24] and 23 in *Paulownia fortunei* [40]. Despite this progress, the comprehensive identification and characterization of SBPs in Tieguanyin remain unexplored, leaving potential candidate SBP genes associated with stress responses unidentified. In the present study, we conduct a systematic identification of the SBP family genes by leveraging genomic data and then proceed to analyze their physicochemical characteristics and gene structures, as well as chromosomal distribution patterns. Meanwhile, the expression trends of its sequence motifs, promoter cis-acting elements, and nine different *CsSBPs* genes under light, dark, and temperature stress aim to establish a reference framework for exploring the role of Tieguanyin SBP genes in stress resistance. This study holds significant importance in elucidating the fundamental molecular mechanisms governing SBP responses to light, shade, and cold stress. It aims to provide vital genes of high quality, which are essential for bolstering Tieguanyin’s resistance to these stressors through genetic modification and breeding initiatives.

## 2. Results

### 2.1. Recognition of Components Belonging to the CsSBP Gene Family in Tieguanyin

In this research, we managed to identify a total of 22 SBP-box genes within the genome of Tieguanyin. These genes were systematically named *CsSBP1* through *CsSBP22* according to their respective phylogenetic relationships (Table 1 and Figure 1). Table 1 provides a thorough description of the *CsSBP* gene family found in Tieguanyin, covering different details of these genes. In particular, the differences in their genome sequence length are indicated. The shortest *CsSBP6* was 143 base pairs (bp), and the longest *CsSBP2* was up to 1092 bp. The genomic sequence of the *CsSBP* gene family was 512 bp long. These data are crucial for mastering the genetic diversity and structure of the Tieguanyin *CsSBP* gene family. The molecular weights of the proteins range from 16,312.92 Da (*CsSBP6*) to 120,280.32 Da (*CsSBP2*). The predicted theoretical pI (isoelectric point) values range from 5.56 to 9.50, with 16 of the CsSBP proteins being alkaline, having a pI value greater than 7.0, and the remaining 6 being acidic. All 22 members of this family exhibited a negative hydrophobicity index, indicating that they are hydrophilic proteins. Twenty-two family members revealed a stability index lower than 40; hence, all were considered stable. Notably, subcellular localization prediction indicated that all members of the Tieguanyin SBP families are localized in the nucleus, with five additional members (*CsSBP9, 10, 12, 16, 18*) also present in the cytoplasm.

### 2.2. Examination of the Structural Organization of CsSBP Genes and Analysis of Conserved Motifs

To further characterize the attributes of the 22-member SBP transcription factor family, we generated a phylogenetic tree by employing protein sequences derived from *CsSBP* (Figure 2A). The Conserved Domain Database (CDD) was consulted for an online search, and TBtools was employed to analyze the conserved motifs, conserved domains, and exon–intron organization. Using the MEME suite, we identified ten conserved motifs. These protein motifs within the *CsSBP* genes are depicted by colorful boxes in the illustration, with each box representing a distinct motif (Figure 2B). Additionally, a functional conserved domain analysis was conducted using the Pfam database (Figure 2C). The majority of family members grouped within the same branch exhibited similar motifs and domain structures, while diversity was observed among the *CsSBP* subfamilies. CsSBP10 and CsSBP16 were grouped, containing a similar combination of motifs, indicating they may perform similar functions. Moreover, the analysis revealed that only CsSBP2 possesses the PRK12323 superfamily domain among all family members. The Ank_2 superfamily domain was exclusively found in CsSBP22. However, the subfamily members, CsSBP11 and CsSBP15, exhibited very similar structures, both containing ANKYR superfamily domains. This finding prompts further investigation into the distinct biological roles of proteins with these unique domains in mind. Through the exon–intron structure analysis, it was determined that one member retained only the coding sequence (CDS) region, while the remaining 21 encompassed both the untranslated region (UTR) and CDS regions (Figure 2D).

### 2.3. Evolutionary Relationship Analysis and Sequence Alignment of SBP Protein Sequences

To elucidate the evolutionary relationships within the CsSBP transcriptional activator family, a comprehensive phylogenetic analysis was conducted using SBP proteins derived from three distinct species: *Arabidopsis thaliana*, *Solanum lycopersicum* L., and Tieguanyin. These species harbored 30, 17, and 22 SBP members, respectively. The 22 SBP proteins were categorized into 6 distinct subgroups (I–VI) (Figure 3). The phylogenetic analysis indicated that subgroup VI harbored the highest number of SBP family members, whereas only six SBP family genes were identified in subgroup II. Among these, subgroup III comprised the most CsSBP family members, with a total of 6 *CsSBP* family members (*CsSBP14, CsSBP5, CsSBP8, CsSBP1, CsSBP7,* and *CsSBP6*), followed by subgroup I (*CsSBP2, CsSBP22, CsSBP11,* and *CsSBP15*) and group V (*CsSBP13, CsSBP9, CsSBP21,* and *CsSBP4*), while there are only three *CsSBPs* in group IV (*CsSBP20, CsSBP3,* and *CsSBP17*) and group VI (*CsSBP19, CsSBP12,* and *CsSBP18*), and two in group II (*CsSBP10* and *CsSBP16*).

Furthermore, SBPs that clustered together possess high similarity in protein sequence, suggesting that they may have similar functions. In our study, SBPs of Tieguanyin generally cluster with those from *Arabidopsis thaliana*.

### 2.4. Chromosomal Distribution and Intraspecific Collinearity Analysis of CsSBP

We carried out a detailed inspection of the chromosomal positions of *CsSBP* genes within the genome of Tieguanyin. Our analysis revealed that the 22 identified *CsSBP* genes are dispersed across 11 distinct chromosomes, exhibiting a seemingly random distribution pattern. Notably, there is no distribution of *CsSBP* genes on four specific chromosomes, suggesting potential selective pressures or functional constraints operating on these genomic regions (Figure 4). Among the chromosomes that include *CsSBP* genes, the number of CsSBP members on Chromosome 05 is the highest with 5 (22.7%), namely *CsSBP 7*, *CsSBP 8*, *CsSBP 9*, *CsSBP 10*, and *CsSBP 11*, indicating a clustering hotspot for this gene family on this particular chromosome, followed by Chromosome 06, which has 3 *CsSBP* genes. There is only one *CsSBP* gene on Chromosome 03 (*CsSBP5*), Chromosome 04 (*CsSBP2*), Chromosome 07 (*CsSBP1*), Chromosome 08 (*CsSBP6*), and Chromosome 11 (*CsSBP22*). The CsSBP genes’ uneven distribution pattern on chromosomes may be attributed to genetic variations that occurred during evolution. We discovered that a mere five *CsSBP* genes exist without corresponding colinear gene pairs, implying they might have undergone unique evolutionary trajectories or experienced rearrangements not shared with their paralogous counterparts. Conversely, the majority of *CsSBP* genes, amounting to 15 instances, form colinear gene pairs, underscoring a significant degree of conservation in gene order.

### 2.5. Prediction Analysis of Cis-Acting Elements for CsSBP Gene Families

Through preliminary experimental validation and data analysis, many functional elements and sites play crucial roles in regulating the diverse activities of the CsSBP protein. To confirm these findings, we conducted several sequence alignments. The results suggest that a majority of the elements within the *CsSBP* gene are implicated in growth and development, hormonal regulation, pathogen resistance, and photoresponsive processes. These elements mainly include the hormone MeJA, light, salicylic acid responsiveness, abscisic acid, anaerobic induction, regulation of zearalenone metabolism, low temperature and light stimulation conditions, MYB binding sites in response to light and drought induction, auxin, gibberellin, part of circadian control conserved DNA modules involved in light response, auxin response, meristem expression, and part of the conserved array of DNA modules (CMA 1) (Figure 5). Partial light response modules were found in a total of 22 (100%) CsSBP promoter sequences, and low-temperature response elements were identified in 8 (36%) CsSBP promoter sequences. Additionally, the promoter sequences of *CsSBP* genes are rich in ABA, MeJA, GA, and SA response elements, suggesting that these genes might have a function in the transcriptional regulation of responses to abiotic stress and hormonal signaling. We hypothesized that these abiotic stresses may regulate the function of CsSBP proteins by affecting the activity or expression levels of relevant elements within the CsSBP gene.

### 2.6. Syntenic Analysis of CsSBPs Genes

To investigate deeper into the evolutionary mechanisms of the *CsSBP* family, we conducted a comparative analysis of collinearity among *CsSBP* gene pairs in the genomes of *Arabidopsis thaliana*, *Solanum lycopersicum* L., and *Oryza sativa* (Figure 6). The results reveal that *CsSBP* established 20 collinear gene pairs with *AtSBP*, 21 with *SiSBP*, and 14 with *OsSBP*.

Several *CsSBP* genes have been recognized as homologs of the individual *AtSBP*, *SISBP*, and *OsSBP* genes. Similarly, multiple *AtSBP*, *SISBP*, and *OsSBP* genes exhibit homology among themselves. Based on these collinearity relationships, we can deduce that the expansion of this gene family likely occurred before the divergence of species such as Tieguanyin, *Arabidopsis thaliana*, and *Solanum lycopersicum* L.

### 2.7. Expression Trends of CsSBP Genes upon Exposure to Light, Shade, and Cold Treatments

In prior research, the significant function of the *SBP* gene family in response to abiotic stress has been well-documented. To confirm the expression patterns of *CsSBPs* under abiotic stress conditions, we employed quantitative real-time polymerase chain reaction (qRT-PCR) to analyze the expression of nine *CsSBPs* belonging to six subgroups (I–VI) during three types of abiotic stress treatments: light, shade, and cold (Figure 7). The results indicated that all *CsSBP* genes showed distinct up and downregulated expression patterns under different stress treatments. Under light stress, most of the *CsSBP* genes’ expression was first increased, then decreased, and then increased, significantly upregulated at 4 h and 8 h, downregulated at 12 h, and upregulated at 24 h. *CsSBP19* demonstrated the most remarkable rise, followed by *CsSBP17*, *CsSBP12*, *CsSBP1*, and *CsSBP8*. Most of the *CsSBPs* reached their peak at 8 h and had a significant decrease at 12 h except *CsSBP12*, *CsSBP17*, and *CsSBP21*, which reached their peak at 24 h. This suggested that the *SBP* family is temporarily downregulated at 12 h and subsequently upregulated under light stress, indicating that the *SBP* family exhibits a favorable response to light stress. Under shade stress, the relative expression levels of genes *CsSBP1*, *CsSBP2*, *CsSBP8*, *CsSBP10*, *CsSBP16*, and *CsSBP19* showed regular changes at different time points. Specifically, these genes were relatively high at control and 12 h and relatively low at 8 h and 24 h. This suggests that under shade, these genes may be upregulated in the beginning (within 12 h) in response to light changes, followed by a decrease in expression between 12 h to 24 h. This trend possibly reflects the specific roles of these genes and regulatory mechanisms in plant adaptation to low-light environments. Under cold stress, the expression of most *CsSBPs* peaked at the 24-h time point, which may indicate that members of the *CsSBP* family possess the potential capacity to withstand cold stress.

## 3. Discussion

Tieguanyin is one of the world’s famous tea varieties, gaining popularity due to its unique aroma, taste, and beneficial health effects. In this context, the *SBP* (Squamosa Promoter Binding Protein) gene family is identified as a vital transcription factor that aids organisms in overcoming various constraints and exerts a remarkable influence on growth and development. This finding provides a new perspective to further study the growth mechanism, improves the quality and yield of Tieguanyin tea, and lays a profound foundation and prospect for future research.

In this study, the Tieguanyin *SBP* gene family was thoroughly examined, resulting in the identification of 22 genes as members of the SBP family that encode transcription factors. Subsequently, the basic biochemical properties of CsSBPs, including their molecular weight (MW), theoretical isoelectric point (pI), instability index, grand average of hydrophilicity, and subcellular localization, were predicted and summarized in Table 1. Compared to SBP families found in other species, *CsSBPs* have universal similarities to SBP families in other species in terms of molecular weight size (16,312.92–120,280.32), protein size (143–1092), and isoelectric points (5.56–9.50). This indicates that, despite potential variations in amino acid sequences and functional characteristics, the overall dimensions of SBP proteins, encompassing CsSBPs, remain relatively uniform across diverse organisms. This uniformity in molecular weight may hint at shared evolutionary histories and conserved structural attributes that are pivotal for their role as transcription factors. CsSBP is predominantly hydrophilic and is found in various locations within the nucleus and cytoplasm. Notably, CsSBPs localized to the nucleus are suspected to play a role in regulating gene expression. Overall, these predictions provide a significant understanding of the physical characteristics of CsSBP proteins. To attain a thorough comprehension of the SBP gene family, we further examined the motifs, domain architecture, and gene structure of CsSBP proteins (Figure 1).

To elucidate the evolutionary connections among *Arabidopsis thaliana*, *Solanum lycopersicum* L., and Tieguanyin, we assembled a phylogenetic tree of SBPs. Based on established classification criteria, these CsSBP proteins were subsequently grouped into six distinct subgroups (Figure 2). The *CsSBP* genes are the most widely distributed in subfamily III, followed by subfamilies I and V, with the fewest members in subfamily II. The results showed that Tieguanyin had 8 fewer SBP members than *Arabidopsis thaliana* and 5 more SBP members than *Solanum lycopersicum* L. Notably, the three SBP family members were distributed in each subfamily, showing a pattern of crossover distribution but not forming isolated lineages (Figure 2). This demonstrates that the SBP family genes of dicot plants currently show no significant change in the evolutionary direction and are still conserved. Since genes clustered in the same subgroup are likely to have similar functional properties, we can infer the potential functions of the CsSBPs that are clustered with the investigated SBPs. *CsSBPs* gather more frequently in Tieguanyin and *Arabidopsis thaliana*. Therefore, we speculate that *CsSBP* genes have similar functions to *Arabidopsis thaliana* and other homologs. In the subgroup, AtSBP has the most members. Interestingly, all *CsSBP* members in the subfamily (*CsSBP12, CsSBP18, CsSBP19*) contain cis-elements in response to low temperatures. *CsSBPs* that are clustered with *Arabidopsis thaliana* may target CsSBP through miR156SBP, promote the translation and expression of C-REPEAT binding factor (CBF) in response to low temperature, and thus improve the plant’s cold tolerance [41,42]. Remarkably, *AtSBP9* expression was significantly induced under cold stress conditions, suggesting a pivotal role for AtSBP9 in enhancing cold tolerance during the plant seedling phase [43]. The SBP-box participates in the regulation of hormonal signaling pathways that have an impact on plant growth and developmental stages. *AtSPL8* aggregates *CsSBP12, CsSBP18,* and *CsSBP19* and mediates *Arabidopsis thaliana* development by affecting gibberellin synthesis [44]. In subgroup I, *AtSBP3* is believed to play a crucial role in response to long-day conditions, photoperiod-induced flowering, and the synergistic FT-FD modules. It is also closely associated with the interpretation of light signals [45]. In addition, we conducted a collinear analysis of Tieguanyin with other dicot and monocot plants (Figure 4). Tieguanyin and *Solanum lycopersicum* L. exhibited the highest number of homologous genes, followed by *Arabidopsis thaliana*. Furthermore, SBP had more collinear gene pairs with dicots than with monocots, which further confirms the close affinity between Tieguanyin and dicots.

Based on the findings from our prior research, we identified and selected nine *CsSBP* genes belonging to various subfamilies for an extensive analysis of their expression patterns. The qRT-PCR technique was employed to evaluate the expression patterns of *CsSBP* genes in response to light stress, shade stress, and cold stress. *CsSBP* was almost uniformly upregulated under multiple stresses, indicating a potential role in enhancing stress resistance. This finding implies that CsSBP functions as a transcriptional activator, regulating plant adaptive mechanisms in response to environmental stressors. Correspondingly, *CsSBP17, CsSBP19,* and *CsSBP21* changed most significantly when encountering the three stresses. These results indicate that the SBP gene family has a significant role in the development of Tieguanyin, facilitating its recovery from diverse environmental stresses. In conclusion, the identification and functional analysis of *CsSBP* under light, shade, and cold stresses provide promising genetic strategies to improve the stress resistance of Tieguanyin.

SBP is a crucial transcription factor that enables plants to manage and adapt to a wide range of stress conditions. In a previous study, we observed that Ca²⁺ stress-activated Ca²⁺ channels, including TPCs, CNGCs, and MS, to transport Ca²⁺ [46]. As the cytoplasmic Ca²⁺ concentration increases, numerous cellular physiological activities are modulated accordingly [47]. Appropriately elevating the Ca²⁺ concentration can boost the calmodulin (CaM) content and enhance the freezing resistance of protoplasts [48]. Our research centered on the alterations in cellular metabolic activity triggered by CsSBP in the context of environmental adaptation. Under the stress of light and cold, there is a significant increase in the reactive oxygen species (ROS) content within plant cells. This excess amount can cause toxicity to plants. Continuous stress causes self-protective mechanisms, and plants can maintain the dynamic balance between free radical production and clearance by increasing their protective enzyme content. That is, using the antioxidant enzyme system, including POD, SOD, and CAT, to remove ROS in chloroplasts, produce antioxidants, and regulate the level of compatible solutes [49,50]. Water loss constitutes a notable feature among cold-sensitive plant species subjected to cold stress. This phenomenon is closely linked to a swift reduction in root water uptake and transport capabilities, which, in turn, triggers a substantial increase in abscisic acid (ABA) biosynthesis within the plant. Elevated levels of ABA are correlated with enhanced cold resistance in plants [51]. CBP60g is a calmodulin-binding protein that interacts with CAM and plays a crucial role in the initial stages of the plant’s defense response [52]. HSP is a heat shock protein produced in plants, especially in reaction to heat stress. However, variations in the quantity of HSP can also be observed in plants subjected to cold stress [53,54]. Under the accumulation of these key proteins, the increased content of plant hormones such as ABA, MeJA, and SA in plants can stimulate the expression of defense genes, which may enhance plant cold tolerance and maintain the good quality of Tieguanyin. Collectively, these results suggest that clearing ROS and preserving osmotic balance are critical for enabling plants to restore growth under stress conditions [55,56]. In our study of cis-elements (Figure 5), we discovered a potential relationship between CsSBP and these key response proteins, implying that *CsSBP* plays an important role in resisting environmental stress, possibly through the interaction or regulation of these proteins (Figure 8). This mechanism is a hypothesis derived from our study of *CsSBP* and other studies of shared plant responses to stress. Additional experiments are required to confirm whether *CsSBP*s indeed assist plants in overcoming stress challenges by regulating the expression of the aforementioned proteins.

## 4. Materials and Methods

### 4.1. Identification and Characterization of the SBP Gene Family in Te TGY Plants

At the Tea Research Institute of Fujian Academy of Agricultural Sciences, PacBio sequencing and de novo genome assembly were used to obtain the genome sequence and annotation data of Tieguanyin plants (TGY). Samples were taken from a Tieguanyin plant in Anxi County, Fujian Province, China. The coordinates are 119.576708° E, 27.215297° N. To identify SBP genes, a search was conducted in the Pfam database (http://pfam.xfam.org/, accessed on 2 July 2024) for the conserved structural domain (PF00447) by employing a hidden Markov model (HMM) analysis. To find genes with these structural domains, Lobster search (v3.0) was utilized. The CsSBP gene family was confirmed by comparing *Arabidopsis thaliana* SBP amino acid sequences in PlantTFDB (accessed on 2 July 2024) with BLASTp from NCBI. Inspecting the structural domains with the NCBI-CDD search tool (https:// www.ncbi.nlm.nih.gov/Structure/bwrpsb/bwrpsb.cgi, accessed on 2 July 2024) and Pfam Web database (http://pfam-legacy.xfam.org/, accessed on 2 July 2024), 22 genes were identified and named CsSBP1-22.

### 4.2. Evolutionary Analysis and Gene Structure of the CsSBP Family 

With a pattern count of 10 and default parameters retained, the MEME website (http://meme-suite.org/ accessed on 2 July 2024) was utilized to identify conserved motifs within the *CsSBP* family. To predict conserved domains, the protein sequences were then uploaded to the NCBI Batch Web CD-Search tool (https://www.ncbi.nlm.nih.gov/, accessed on 3 July 2024). Using the Gene Structure Display Server (https://gsds.gao-lab.org/Gsds_help.php accessed on 2 July 2024), the gene structure, including introns and exons, was visually represented.

### 4.3. Physicochemical Characteristics and Subcellular Localization

Through the ProParam tool on the ExPASy platform (https://www.expasy.org/ accessed on 2 July 2024), the physicochemical attributes of the CsSBP family proteins were examined on 2 July 2024. On the same day, the Plant-mPLoc server (http://www.csbio.sjtu.edu.cn/bioinf/plant-multi/ accessed on 2 July 2024) was additionally employed to predict the subcellular destinations of these proteins.

### 4.4. Phylogenetic Analysis

The protein sequences of SBPs (sequence-binding proteins) in *Arabidopsis thaliana* and *Solanum lycopersicum* L. were obtained from the PlantTFDB database (https://planttfdb.gao-lab.org/, accessed on 2 July 2024). MEGA software, version 11, was used to construct a phylogenetic tree. In this regard, the relevant sequences were initially aligned using ClustalW; thereafter, the maximum likelihood method was applied, using the JTT + G model and 1000 bootstrap replicates for robustness, using the iTOL online tool (https://itol.embl.de/ accessed on 2 July 2024) to visually improve and represent the tree.

### 4.5. Collinearity and Repetition Analysis

The homology between CsSBP genes and SBP genes in Arabidopsis, tomato, tobacco, wheat, rice, and maize was investigated using MCScanX (https://github.com/wyp1125/MCScanX/ accessed on 2 July 2024). Within the CsSBP gene family, segmental and tandem duplication events were identified, and TBtools-II v2.136 was used for visualization of the results.

### 4.6. Non-Biological Stress Treatments

Uniform-sized Te TGY seedlings were split into two groups: 30 seedlings for the control group and 3 seedlings for the stress treatment group. Simulated light (intense light exposure), shade (no light), and cold (4 °C) stress treatments were applied, maintaining a 75% relative humidity. Post-treatment samples were collected at 0, 4, 8, 12, and 24 h. The samples were immediately placed in liquid nitrogen and kept at −80 °C for successive RNA extraction.

### 4.7. Extraction and Quantitative Analysis of RNA

An RNA extraction kit supplied by Omega Bio-Tek (Shanghai, China) was employed to extract RNA from both control and stressed samples. For cDNA synthesis, the EasyScript one-step gDNA removal and cDNA synthesis SuperMix, provided by Transgen (Beijing, China), was subsequently utilized. TransStart Top Green qPCR SuperMix from Transgen (Beijing, China) was used for quantitative real-time polymerase chain reaction (qRT-PCR), following the manufacturer’s instructions for all steps. The reaction mixture for qRT-PCR included 1 μL of cDNA, 2 μL of specific primers, 7 μL of nuclease-free water (ddH2O), and 10 μL of SYBR Premix Ex TaqTM II. The qRT-PCR thermal cycling protocol included an initial 30-s denaturation at 95 °C, followed by 40 cycles of 5-s denaturation at 95 °C and 30-s annealing/extension at 60 °C. A melt curve analysis concluded the reaction, which involved heating to 95 °C for 5 s, cooling to 60 °C for 60 s, and then cooling to 50 °C for 30 s. Statistical analysis was conducted using GraphPad Prism 9.0 software (available at https://www.graphpad.com/ accessed on 2 July 2024), while the relative gene expression levels were calculated by means of the 2^−ΔΔCT^ method. Ensuring reproducibility in qRT-PCR experiments involved conducting each with three biological replicates, with each replicate having three technical replicates.

### 4.8. RNA Extraction and Statistical Analysis

Total RNA was extracted from both control and stressed samples with the assistance of the RNA Extraction Kit provided by Omega Bio-Tek (Shanghai, China). Following the manufacturer’s guidelines, cDNA synthesis was performed with EasyScript one-step gDNA removal and cDNA synthesis SuperMix. Quantitative RT-PCR was then carried out with the TransStart Top Green qPCR SuperMix from Transgen (Beijing, China). The qRT-PCR reaction mixture contained 1 µL of cDNA, 2 µL of specific primers, 7 µL of ddH2O, and 10 µL of SYBR Premix Ex TaqTM II. The qRT-PCR process consisted of an initial 30-s denaturation step at 95 °C, followed by 40 cycles of 5-s denaturation at 95 °C and 30-s annealing/extension at 60 °C, ending with a melt curve analysis of 5 s at 95 °C, 60 s at 60 °C, and 30 s at 50 °C. Relative to the control, the 2^−∆∆CT^ method was utilized to calculate the expression levels of the CsBES1 genes. For further statistical analysis, GraphPad Prism 9.0 software included unidirectional scatter plot analysis, and Duncan’s multiple range test (accessible at https://www.graphpad.com/ accessed on 2 July 2024) was applied. All quantitative PCR experiments were conducted with three technical replicates per biological replicate, totaling three biological replicates, for the purpose of guaranteeing the reliability of the results.

## 5. Conclusions

This study identified nine *CsSBP* genes in Tieguanyin tea, analyzing their properties, relationships, structures, functions, and expression patterns. Among these, *CsSBP1*, *CsSBP12*, *CsSBP17*, *CsSBP19*, and *CsSBP21*, which contain promoter regions, have a higher probability of playing critical roles in the response to diverse abiotic stresses. The SBP transcription factors may also interact with other signal transduction components, such as CBP60g and HSP, contributing to the regulation of plant growth and development. Further analysis revealed that *CsSBP17* and *CsSBP19* exhibited the most significant expression changes under three types of stress. Specifically, *CsSBP17* was highly upregulated under light and cold stress, suggesting a potential role in light and cold stress resistance, while *CsSBP19* showed the highest upregulation under shade stress, indicating a possible link to shade stress resistance. However, the mechanisms through which *CsSBP* transcription factors regulate Tieguanyin growth and development require further investigation. This study provides a comprehensive analysis of *CsSBP* genes in response to abiotic stress, offering valuable insights into their roles in biological stress responses.

## Figures and Tables

**Figure 1 plants-14-00422-f001:**
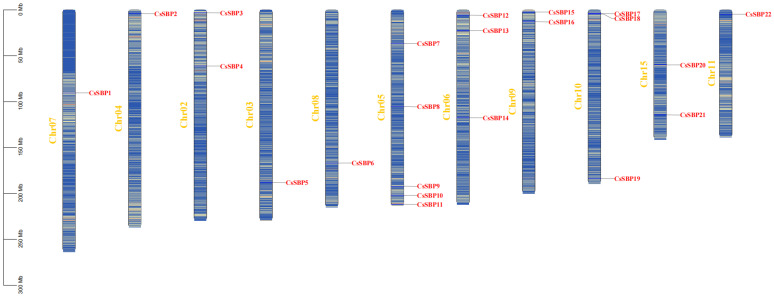
Distribution of *CsSBP* genes in *Camellia sinensis* cv. *Tie-guanyin* chromosomes. In the figure, the blue regions within the chromosomes indicate areas with low gene density, meaning that the number of genes in these regions (each stripe representing approximately 10,000 bp) is relatively low. The yellow regions indicate areas with high gene density, where the number of genes is higher. The white regions indicate the absence of genes. The chromosome sequence number is shown on the left of each chromosome, with a ratio provided on the far left to assess chromosome length and gene position.

**Figure 2 plants-14-00422-f002:**
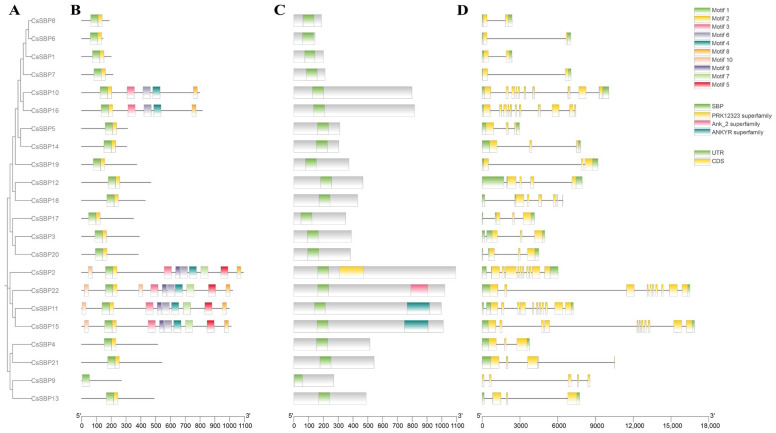
Protein motifs, domain composition and structures of CsSBP gene family in Tieguanyin. (**A**) A phylogenetic tree was constructed in MEGA using the maximum likelihood algorithm with a bootstrap value of 1000. (**B**) The colorful boxes represent distinct motifs within the protein sequences of CsSBP genes. (**C**) Analysis of functional conserved domains was performed in the Pfam database. (**D**) The gene organization of the *CsSBP* family is illustrated, where the coding sequence (CDS) is represented by yellow rectangles and the untranslated region (UTR) by green rectangles. Introns are denoted by black lines.

**Figure 3 plants-14-00422-f003:**
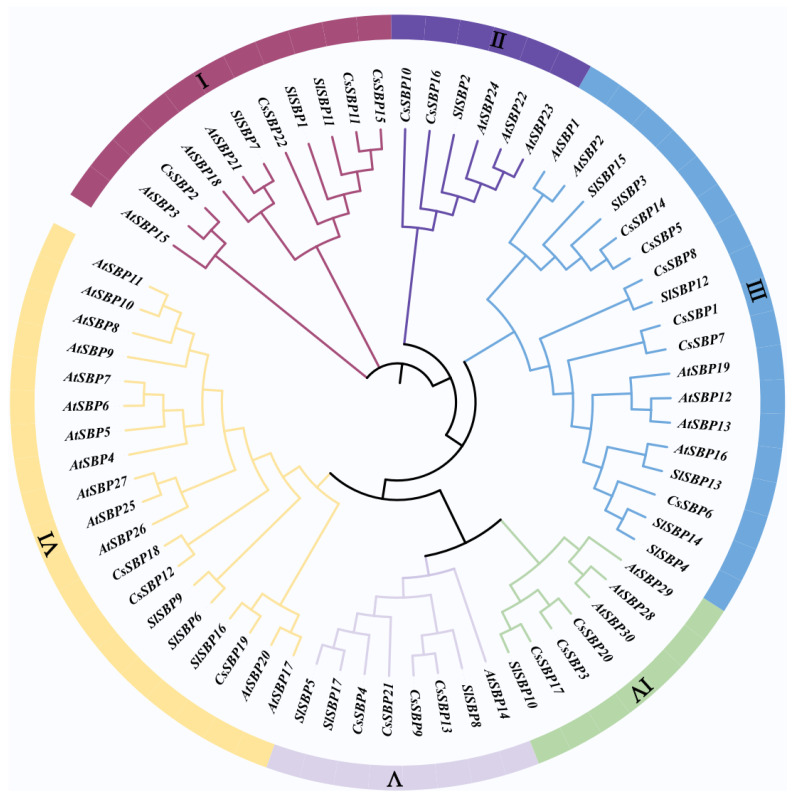
The phylogenetic analysis of SBP proteins originating from Tieguanyin (CsSBP), *Arabidopsis thaliana* (AtSBP), and *Solanum lycopersicum* L. (SISBP) was executed through the utilization of the neighbor-joining approach. In addition, the maximal likelihood method was engaged and the bootstrap value was established as 1000. The six subgroups of SBP proteins (groups I–VI) are distinguished by unique colors in the outermost circle.

**Figure 4 plants-14-00422-f004:**
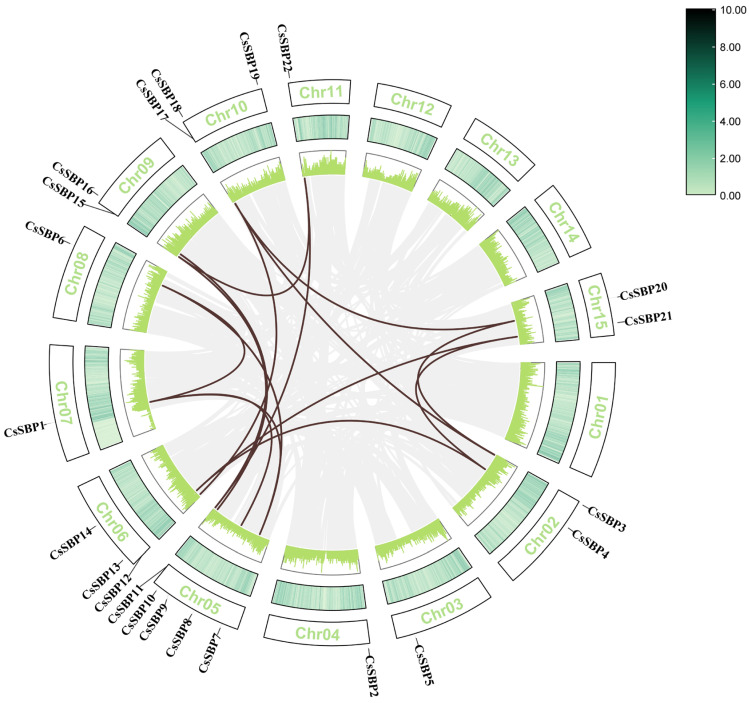
In the analysis of synteny for the *CsSBP* family in Tieguanyin, gray lines denote all regions of synteny within the Tieguanyin genome, whereas brown lines signify pairs of duplicated *CsSBP* genes. The number corresponding to each chromosome is displayed in a rectangular box.

**Figure 5 plants-14-00422-f005:**
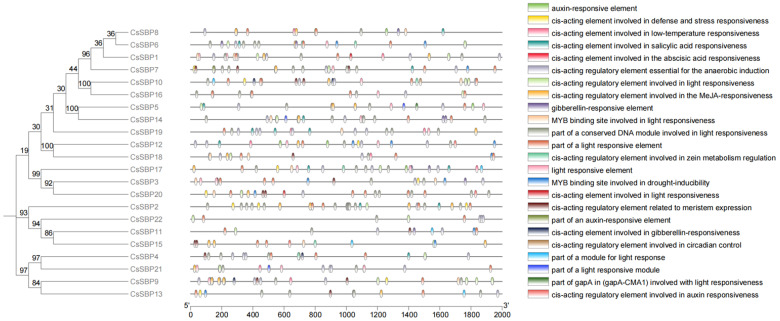
Predicted cis-acting regulatory elements within the promoter sequences of the *CsSBP* genes are shown. On the left, the phylogenetic tree with branches marked by bootstrap values is illustrated. The promoter location at −2000 bp is exhibited on the right. The cis-acting regulatory elements within this promoter region are classified into 24 unique types, each denoted by a distinct color. The bottom axis indicates the abundance of each type of cis-acting element.

**Figure 6 plants-14-00422-f006:**
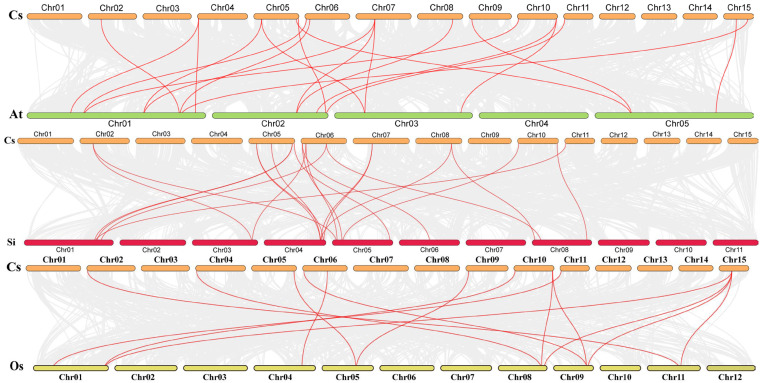
*Arabidopsis thaliana, Solanum lycopersicum* L., *Oryza sativa,* and Tieguanyin *SBP* gene synteny analysis. The red lines emphasize the syntenic *SBP* gene pairs, whereas the gray lines in the background depict the collinear blocks within the genomes of Tieguanyin in comparison to other plants.

**Figure 7 plants-14-00422-f007:**
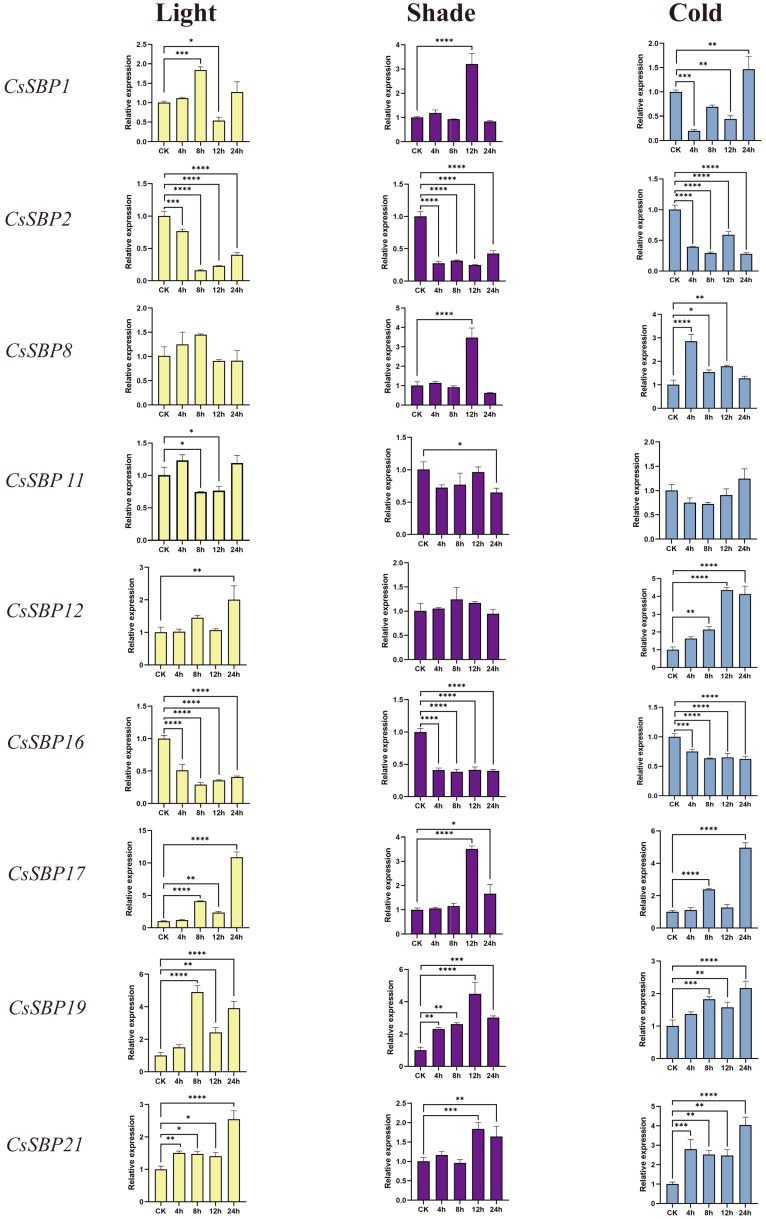
The expression patterns of the candidate CsSBP genes were examined under various stress conditions, with error bars indicating the standard deviation (SD). One-way ANOVA was used for statistical analysis to ascertain significant differences, with the number of asterisks indicating the level of significance as follows: * for *p* ≤ 0.05, ** for *p* ≤ 0.005, *** for *p* ≤ 0.0005, and **** for *p* ≤ 0.0001.

**Figure 8 plants-14-00422-f008:**
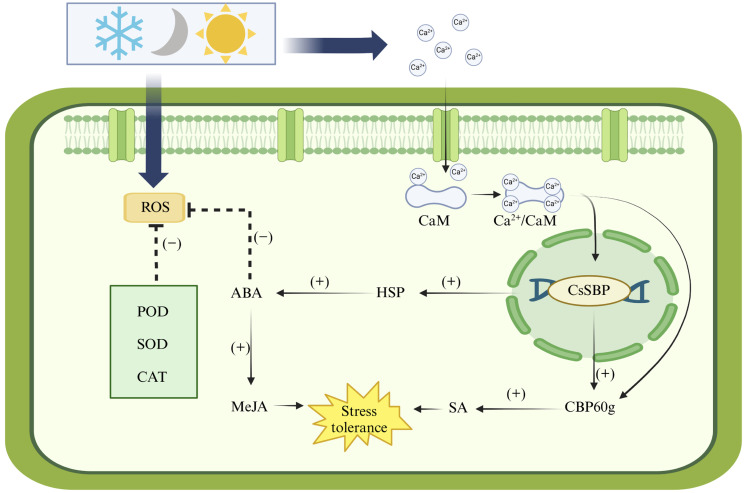
Predicted mechanisms of SBPs that enable plants to withstand intense environmental stresses.

**Table 1 plants-14-00422-t001:** Sequence characteristics of the 22 *CsSBP* genes identified in Tieguanyin.

Name	Sequence ID	Number of Amino Acid	Molecular Weight	Theoretical pI	Instability Index	Grand Average of Hydropathicity	Predicted Location(s)
CsSBP1	CsTGY07G0000556	199	21,654.11	8.94	73.91	−0.992	Nucleus
CsSBP2	CsTGY04G0000059	1092	120,280.32	8.25	56.07	−0.463	Nucleus
CsSBP3	CsTGY02G0000059	388	43,086.26	8.88	53.74	−0.622	Nucleus
CsSBP4	CsTGY02G0001323	513	56,414.44	7.63	45.87	−0.607	Nucleus
CsSBP5	CsTGY03G0002650	309	34,415.78	8.84	69.12	−0.851	Nucleus
CsSBP6	CsTGY08G0001893	143	16,312.92	7.00	80.77	−1.362	Nucleus
CsSBP7	CsTGY05G0000639	210	23,366.81	9.17	73.25	−1.253	Nucleus
CsSBP8	CsTGY05G0001480	185	20,895.09	9.05	47.50	−1.221	Nucleus
CsSBP9	CsTGY05G0002407	268	30,638.66	9.06	45.60	−0.104	Cytoplasm Nucleus
CsSBP10	CsTGY05G0002587	797	89,588.18	5.80	58.33	−0.314	Cytoplasm Nucleus
CsSBP11	CsTGY05G0002815	995	110,008.4	6.33	44.60	−0.393	Nucleus
CsSBP12	CsTGY06G0000121	466	51,122.98	8.45	56.65	−0.685	Cytoplasm Nucleus
CsSBP13	CsTGY06G0000542	488	54,137.89	7.61	49.82	−0.505	Nucleus
CsSBP14	CsTGY06G0002194	303	34,064.75	9.50	63.65	−0.762	Nucleus
CsSBP15	CsTGY09G0000082	1009	111,997.33	6.52	48.90	−0.314	Nucleus
CsSBP16	CsTGY09G0000361	814	90,960.3	6.29	53.05	−0.299	Cytoplasm Nucleus
CsSBP17	CsTGY10G0000081	349	38,204.28	7.64	59.34	−0.696	Nucleus
CsSBP18	CsTGY10G0000091	429	47,157.92	8.87	54.32	−0.605	Cytoplasm Nucleus
CsSBP19	CsTGY10G0002410	371	39,680.64	8.59	48.25	−0.679	Nucleus
CsSBP20	CsTGY15G0000933	381	41,795.4	7.19	65.80	−0.622	Nucleus
CsSBP21	CsTGY15G0001546	541	58,874.01	6.53	43.27	−0.650	Nucleus
CsSBP22	CsTGY11G0000061	1019	112,946.93	5.56	54.95	−0.363	Nucleus

## Data Availability

Data available in a publicly accessible repository. The original data presented in the study are openly available in the National Center for Biotechnology Information under accession number JAFLEL000000000 and in the GWH (https://bigd.big.ac.cn/gwh/, accessed on 2 July 2024) under the accession numbers GWHASIV00000000.

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
