# Peer review of "Phylogenetic and Expression Analysis of SBP-Box Gene Family to Enhance Environmental Resilience and Productivity in Camellia sinensis cv. Tie-guanyin"

_plants, 2025, doi:10.3390/plants14030422_

Round 1

Reviewer 1 Report

Comments and Suggestions for Authors

The MS conducted a comprehensive identification of SBP gene family from Tieguangyin (Camellia sinensis var. Tieguanyin) genome, analyzing physicochemical characteristics, gene structures, and chromosomal distribution patterns of these genes, and monitored the transcript level of nine CsSBP genes under light, dark and cold treatment. Although authors had discussed the potential role of these CsSBP genes under abiotic stresses, it is too superficial. I have some concerns about your research. First, how can you identify the effect of intense light exposure, darkness or cold stress on C. sinensis seedlings. You should monitor some physiological parameters to identify the effect of those abiotic stress on seedlings, such as the ROS level and its associated enzyme level. Second, it lacks depth only using the CsSBP transcripts. You should integrate some physiological index with the CsSBP transcripts and discuss them more comprehensively. Moreover, SBP as the TFs, you should monitor the expression of some known downstream genes to identify the potential role of CsSBP genes under the three abiotic stresses. In summary, it cannot be accepted as its current state.

Comments on the Quality of English Language

1.     Some latin names of plant species are not italicized.

2.     Many sentences contain unnecessary periods.

3.     “light, dark, and cold conditions”, “light, dark and temperature stress”, “Light, Shade and Cold Treatments” and “Non-Biological Stress Treatments”, these inconsistencies in expressions are existed in MS.

Reviewer 2 Report

Comments and Suggestions for Authors

Reviewer suggestions and thoughts basing in a research article “Phylogenetic and Expression Analysis of SBP-box Gene Family to Enhance Environmental Resilience and Productivity in Ca-mellia sinensis cv. Tie-guanyin”

Quality of English is not good enough.The abstract is clear and well organized, but a very few improvements can make it even stronger. Here are review with suggestions:

Abstract

In this papers title, ' Tie-guanyin ' is used, but in the abstract it is' Tieguanyin '. It is suggested to revise it to be consistent. Repeat the two sentences in lines 20 and 59.

Introduction

Background for the research is good. However, there are some areas where bit improvements could be made in terms of flow.

Introduction Whether the first and second paragraphs are placed behind the third paragraph?

It is recommended to add some new literature to explain the latest progress in your research direction, as there are many research achievements in this area, which can help readers understand the latest developments.

The names of genes should be highlighted in italics. However, in the introduction and other parts of this article, there are many genes that are not italicized. It is recommended to make corrections, such as the "CsSBP" in lines 88, as well as other places that will not be listed here.

Result

1. Line 248-259: Can you change the mentioned plant name to the Latin name?

2. Line 162:Authors need to define all the abbreviations in the first place as they appear. Check throughout the MS.

Discussion

There are far too many commonly known statements in the discussion section.Overall, the paper provide an excellent, in-depth analysis of the SBP-box Gene family in response to abiotic stresses.

Conclusion

Overall, the conclusion is strong and effectively summarize the key findings of your study.

References

check as some journal titles are all uppercase and many are lowercase.Some journals are abbreviated, and some are not. Be consistent for all the references.
